# Comparative Analysis of Statistical and Machine Learning Techniques for Rice Yield Forecasting for Chhattisgarh, India

**Anurag Satpathi** [1] , **Parul Setiya** [1], **Bappa Das** [2] , **Ajeet Singh Nain** [1] , **Prakash Kumar Jha** [3,*] , **Surendra Singh** [4] and **Shikha Singh** [5]

1    Department of Agrometeorology, College of Agriculture, G.B. Pant University of Agriculture and Technology, Pantnagar 263153, India
2    ICAR Central Coastal Agricultural Research Institute, Old Goa 403402, India
3    Sustainable Intensification Innovation Lab, Kansas State University, Manhattan, KS 66506, USA
4    Columbia Basin Agricultural Research Center, Oregon State University, Adams, OR 97810, USA
5    Hermiston Agricultural Research and Extension Center, Oregon State University, Hermiston, OR 97838, USA
*    Correspondence: pjha@ksu.edu

**Abstract:** Crop yield forecasting before harvesting is critical for the creation, implementation, and optimization of policies related to food safety as well as for agro-product storage and marketing. Crop growth and development are influenced by the weather. Therefore, models using weather variables can provide reliable predictions of crop yields. It can be tough to select the best crop production forecasting model. Therefore, in this study, five alternative models, viz., stepwise multiple linear regression (SMLR), an artificial neural network (ANN), the least absolute shrinkage and selection operator (LASSO), an elastic net (ELNET), and ridge regression, were compared in order to discover the best model for rice yield prediction. The outputs from individual models were used to build ensemble models using the generalized linear model (GLM), random forest (RF), cubist and ELNET methods. For the previous 21 years, historical rice yield statistics and meteorological data were collected for three districts under three separate agro-climatic zones of Chhattisgarh, viz., Raipur in the Chhattisgarh plains, Surguja in the northern hills, and Bastar in the southern plateau. The models were calibrated using 80% of these datasets, and the remaining 20% was used for the validation of models. The present study concluded that for rice crop yield forecasting, the performance of the ANN was good for the Raipur ($R^2_{cal}$ = 1, $R^2_{val}$ = 1 and $RMSE_{cal}$ = 0.002, $RMSE_{val}$ = 0.003) and Surguja ($R^2_{cal}$ = 1, $R^2_{val}$ = 0.99 and $RMSE_{cal}$ = 0.004, $RMSE_{val}$ = 0.214) districts as compared to the other models, whereas for Bastar, ELNET ($R^2_{cal}$ = 90, $R^2_{val}$ = 0.48) and LASSO ($R^2_{cal}$ = 93, $R^2_{val}$ = 0.568) performed better. The performance of the ensemble model was better compared to the individual models. For Raipur and Surguja, the performance of all the ensemble methods was comparable, whereas for Bastar, random forest (RF) performed better, with $R^2$ = 0.85 and 0.81 for calibration and validation, respectively, as compared to the GLM, cubist, and ELNET approach.

**Keywords:** yield forecasting; SMLR; ANN; LASSO; ELNET; ridge regression

## 1. Introduction

Global food production needs to double by 2050 in order to meet the demand of the rapidly growing population [1,2]. On the other hand, the current yield of growth rates for the major cereals that are grown across the globe are not high enough to meet this target [3]. Environmental changes, particularly global warming and climate variability, are key concerns that have a negative impact on agriculture [4]. This may result in a decline in crop production [5], making the world more food insecure. As the global population is projected to reach 9 billion people by 2050, governments all over the world need to be well prepared to deal with supply shocks of major cereals.

The FAO (Food and Agriculture Organization) reports that the demand for and consumption of grains have grown significantly relative to production in developing nations

such as India. From 1964 to 2030, there will have been an increase in demand for rice, wheat, and other coarse grains [6]. Cereal imports in developing countries increased significantly to meet the rising demand, growing from 39 million tons annually in 1970 to 130 million tons annually by 1997–1999. The increase in imports is anticipated to continue and may be accelerated in the upcoming years. These developing nations are predicted to import 265 million tons of grains by 2030, which is 14% of their total yearly consumption [6]. Nations that do not consider taking action to lessen their overall reliance on imports for conventional crops could suffer greatly as a result of these conditions. Therefore, it is a global challenge to alter the current situation in the future and make nations more and more self-sufficient in meeting their food demands, which in turn requires accurate and timely forecasting of crop yields. Crop yield prediction is one of the most difficult tasks in precision agriculture. The ability to forecast crop yields enables the relevant authorities to make appropriate decisions to ensure food security. In addition to soil, genotype, and management techniques, weather conditions have a significant impact on crop yield [7]. Around 30% of annual production is lost due to unfavorable weather conditions all over the world [8]. As a result, there is a significant demand for models that provide accurate yield predictions before a harvest, which may be utilized by the government, policymakers, and farmers to plan ahead of time.

One of the most important and difficult jobs in applied sciences is future prediction. It requires computational and statistical approaches for identifying relationships between past and short-term future values, as well as adequate strategies to deal with longer horizons, in order to create effective predictors from historical data [9]. Every aspect of modern life is being improved by the incorporation of machine learning, including marketing software, equipment maintenance, health-monitoring systems, crop yield prediction, and the study of soil. For example, Raja and Shukla [10,11] employed an extreme learning machine (ELM) and hybrid grey wolf-optimized artificial neural network models to obtain a more realistic prediction of the ultimate bearing capacity and settlement of a geosynthetic-reinforced sandy soil. Machine learning is adding intelligence to the newest generation of items almost everywhere we look [12].

Traditionally, a crop-cutting experiment was used to measure crop production. However, this takes a long time and requires more human work. Crop yield estimation via crop yield models, which may be constructed using multiple statistical techniques, is another alternative to this old method. Presently, prediction of crop yields using artificial neural networks (ANN), the least absolute shrinkage and selection operator (LASSO), and elastic net (ELNET) is receiving a lot of attention using the relationship of crops to weather datasets [13–17]. Das et al. [14] developed multiple rice yield forecast models for the fourteen different districts on the west coast based on weekly weather indices using LASSO, SMLR, principal component analysis combined with SMLR (PCA-SMLR), ELNET, PCA-ANN, and ANN. Singh et al. [7] used the SMLR technique to develop a wheat yield forecast model based on weekly weather indices and yield records for the Amritsar, Bhatinda, and Ludhiana districts of Punjab. Based on a dataset of 40 farms in Canterbury, New Zealand, Safa et al. [18] developed an ANN model for wheat yield production. Sridhara et al. [19] used the LASSO, ENET, PCA, ANN, and SMLR techniques to forecast the sorghum crop yield at the district level. The researchers discovered that the constructed ANN model could accurately estimate the wheat yield.

Unfortunately, up to today, little scientific effort has been made to develop the yield forecast model using machine learning techniques for the Chhattisgarh region. The majority of research so far relies on predictions based on traditional statistical models. Therefore, in the present research, an attempt has been made to provide a pre-harvest forecast of the rice crop for the Raipur, Surguja, and Bastar districts of Chhattisgarh using SMLR, ANN, LASSO, ELNET, and ridge regression, and a comparison was made among these techniques to select the best model that can be used to provide rice crop yield forecasts for the districts of Chhattisgarh.

## 2. Materials and Methods

Among the districts of Chhattisgarh, three major districts representing three agro-climatic zones were considered in this study, i.e., Raipur (21°15′ N, 81°37′ E) from the Chhattisgarh plains, Surguja (23°6′ N, 83°11′ E) from the northern hills, and Bastar (19°4′ N, 82°1′ E) from the southern plateau (Figure 1).

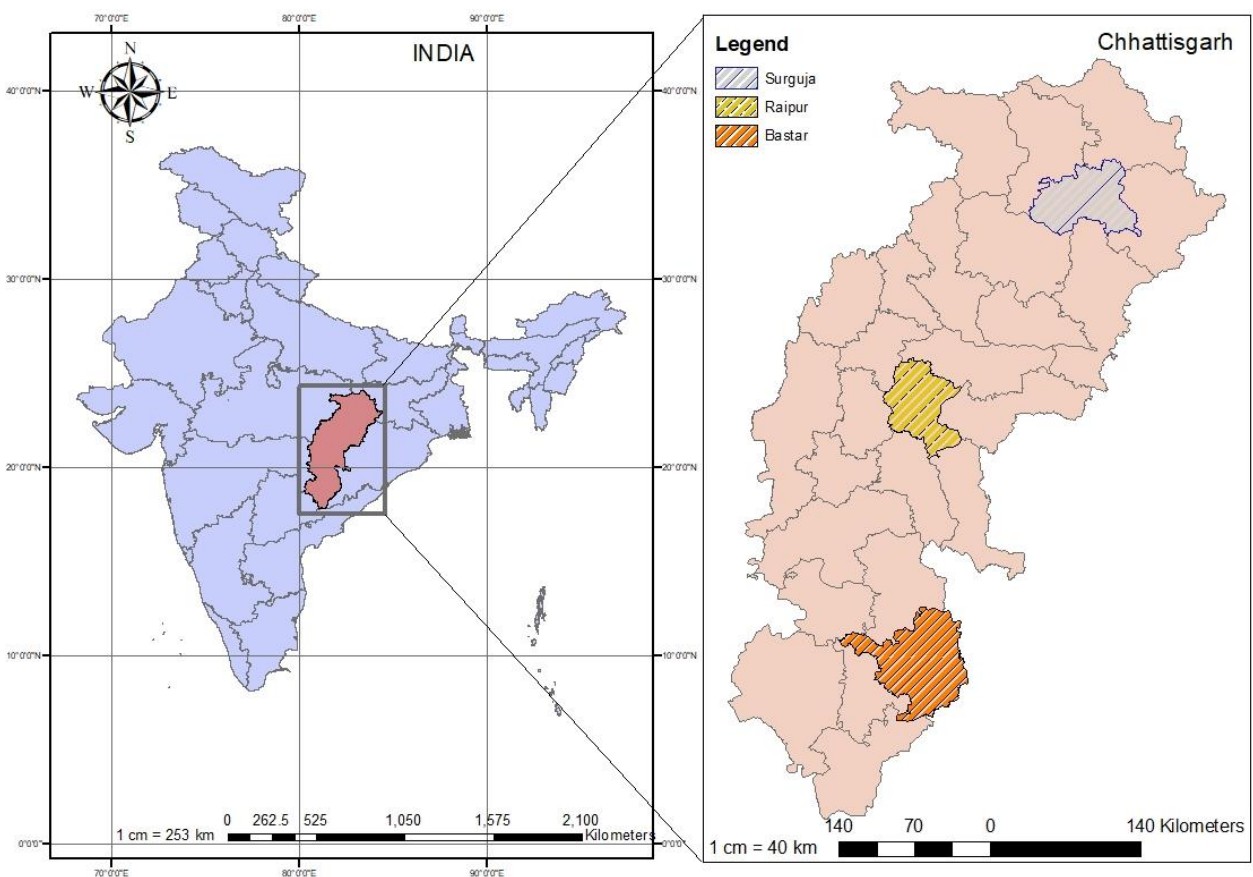

**Figure 1.** Location map of the study area consisting of different districts in Chhattisgarh.

Weather data included maximum temperature (Tmax), minimum temperature (Tmin), rainfall, relative humidity at 7:20 am (RH I) and 2:20 pm (RH II), and sunshine hours of twenty-one years (1998–2018) for all three districts, which were taken from the local observatory situated at the respective districts. The crop yield data for these districts was taken from the Directorate of Economics and Statistics, the Ministry of Agriculture, and Farmers Welfare [20]. Out of the total dataset, 80% of the data was used for calibration, and the remaining 20% was used for the validation of the developed models.

We used the district-level yield data, for which biophysical factors and farming practices are not uniform all over the region, and it is very hard to track the farming practice of each and every field of the districts. Hence, in the present study, we assume that in such a large area as a district, these factors will be either constant, or spatio-temporal variabilities are nullified by each other. For example, if the sowing of a crop in x region of a district is delayed and of y region is advanced, each will neutralize each other's effect/impact on district-level crop productivity. Therefore, inter-seasonal variability in crop yield at the district level is largely influenced by the weather variables. The yield data were detrended prior to data analysis, since it is possible that climatic variability as well as technology differences can affect the trend in the yield data over the long term. However, time is also included as an independent variable in this study. It would be pertinent to mention that time is an important factor in deciding yield at the district level or in a large geographical region due to the fact that time represents cumulative technological advancement (includ-

ing improvement in variety, machinery, disease, insect and weed control measures, etc.) and is a gradual and forward-moving phenomenon. Therefore, considering time as an independent parameter is logical and justifiable. To evaluate the importance of time during model development, these models were redeveloped using only time as an independent variable and without using time as an independent variable. The daily weather data were used to generate the weekly averages. The weighted and unweighted weather indices were then calculated using these average values with the help of formulas provided by Das et al. [13]:

Unweighted weather indices:

$$Z_{ij} = \sum_{w=1}^{n} X_{iw}, \ Z_{ii\prime j} = \sum_{w=1}^{n} X_{iw} X_{i\prime w}, \tag{1}$$

Weighted weather indices:

$$Z_{ij} = \sum_{w=1}^{n} r_{iw}^{j} X_{iw}, \ Z_{ii\prime j} = \sum_{w=1}^{n} r_{ii\prime w}^{j} X_{iw} X_{i\prime w}, \tag{2}$$

Here, $Z$ represents the weather index, $n$ is the week of the forecast, $X_{iw}/X_{i\prime w}$ is the value of the $i$th/$i\prime$th weather variable, the value of $j$ is 0 for all unweighted indices and 1 for all weighted indices, and $r_{iw}^{j}/r_{ii\prime w}^{j}$ is the value of the correlation coefficient of the detrended yield with the $i$th weather variable/ product of the $i$th and $i\prime$th weather variables in the $w$th week.

By following the above procedure, 42 weather indices were generated, as shown in Table 1. The steps involved in model development are shown in Figure 2 [14].

**Table 1.** Weighted and unweighted weather indices for development of multivariate models.

| Parameter | Unweighted Weather Indices | Weighted Weather Indices |
|---|---|---|
| Tmax [1] | $Z_{10}$ | $Z_{11}$ |
| Tmin [2] | $Z_{20}$ | $Z_{21}$ |
| Rainfall | $Z_{30}$ | $Z_{31}$ |
| Sunshine Hour | $Z_{40}$ | $Z_{41}$ |
| Relative Humidity I | $Z_{50}$ | $Z_{51}$ |
| Relative Humidity II | $Z_{60}$ | $Z_{61}$ |
| Tmax × Tmin | $Z_{120}$ | $Z_{121}$ |
| Tmax × Rainfall | $Z_{130}$ | $Z_{131}$ |
| Tmax × Sunshine Hour | $Z_{140}$ | $Z_{141}$ |
| Tmax × Relative Humidity I | $Z_{150}$ | $Z_{151}$ |
| Tmax × Relative Humidity II | $Z_{160}$ | $Z_{161}$ |
| Tmin × Rainfall | $Z_{230}$ | $Z_{231}$ |
| Tmin × Sunshine Hour | $Z_{240}$ | $Z_{241}$ |
| Tmin × Relative Humidity I | $Z_{250}$ | $Z_{251}$ |
| Tmin × Relative Humidity II | $Z_{260}$ | $Z_{261}$ |
| Rainfall × Sunshine Hour | $Z_{340}$ | $Z_{341}$ |
| Rainfall × Relative Humidity I | $Z_{350}$ | $Z_{351}$ |
| Rainfall × Relative Humidity II | $Z_{360}$ | $Z_{361}$ |
| Sunshine Hour × Relative Humidity I | $Z_{450}$ | $Z_{451}$ |
| Sunshine Hour × Relative Humidity II | $Z_{460}$ | $Z_{461}$ |
| Relative Humidity I × Relative Humidity II | $Z_{560}$ | $Z_{561}$ |

[1] Tmax = maximum temperature, [2] Tmin = minimum temperature.

### 2.1. Multivariate Techniques Involved in Model Development

In the present study, we employed statistical (stepwise multiple linear regression (SMLR) and machine learning techniques, such as penalized regression (LASSO, ELNET, ridge regression) and an artificial neural network (ANN). The SMLR is a conventional standard regression model, which is compared with other advanced approaches, and the penalized models address some of the limitations of standard regression for high-dimensional data. LASSO, ELENT, and ridge regression provide shrinkage, which enforces

sparsity, that is, many of the parameter estimates are shrunk to exactly zero. Sparse models have several advantages over traditional regression, such as reduced overfitting (which improves prediction), the accommodation of multicollinearity, and a better ability to fit. They can also be used for variable reduction, where a zero-parameter estimate indicates that the variable is not an important predictor. Another model that we used is the ANN, which is a well-documented AI model inspired by the framework of biological human neurons. It has been successfully applied to numerous problems in different disciplines. In essence, it is a powerful tool for finding a relationship between independent and dependent variables. Hence, in addition to the statistical and shrinkage model, we also included the ANN to establish the association of weather variables with crop yield.

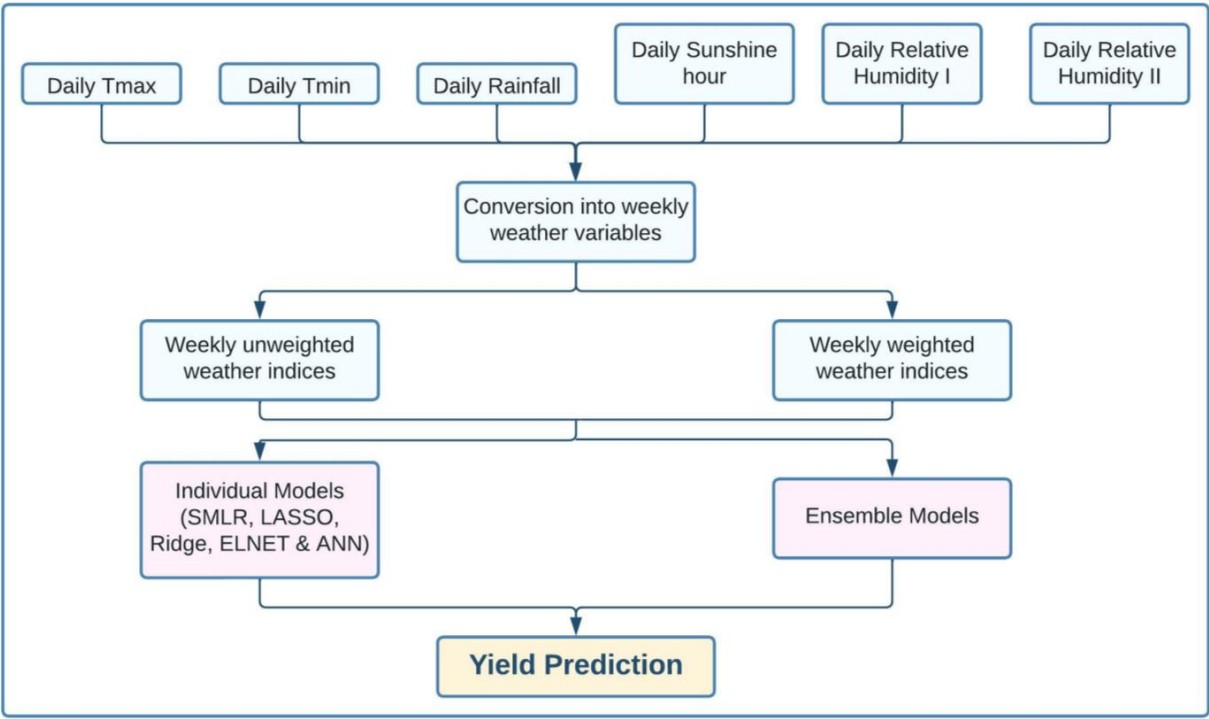

**Figure 2.** Flowchart involving the phases of model development.

Stepwise Multiple Linear Regression (SMLR)

The simplest technique to generate a yield forecast based on a dataset of yield and weather parameters is stepwise multiple linear regression (SMLR). This strategy helps to select the best predictors from a huge number of predictors through a series of automated steps [7,14]. At every stage, the significance of the new variable included in the subsequent step is examined, typically using the *t* statistics and *p*-value.

### 2.2. Shrinkage Regression Models (LASSO, ELNET, and Ridge Regression)

In a situation in which the number of variables in a dataset exceeds the number of samples, the standard linear model typically performs poorly. A more effective alternative to this situation is penalized regression, in which the number of variables in the model is penalized by adding a constraint to the equation. The process of reducing the original size is also known as shrinkage or regularization. The regularization process permits the coefficients of the less important variables to be near to or equal to zero. The penalized regression methods considered in the present study include ridge regression, LASSO, and ELNET.

### 2.2.1. Least Absolute Shrinkage and Selection Operator (LASSO)

The LASSO technique shrinks the coefficients of correlated terms to zero, which ensures that correlated features are included when developing the data-driven model, in

order to prevent overfitting and ensure generalization. In LASSO modeling, correlated features are penalized by reducing their coefficients to zero. The objective function that is minimized by the LASSO algorithm is expressed as [21]:

$$L_{lasso}(\hat{\beta}) = \sum_{i=1}^{n}(y_i - x_i'\hat{\beta})^2 + \lambda \sum_{j=1}^{m}|\hat{\beta}_j|. \tag{3}$$

where $\beta$ is the regression coefficient associated with the input parameters of the LASSO model; $x$ and $y$ are the input and output, respectively, $n$ is the number of samples in the training dataset, and the hyper-parameter $\lambda$ is the penalty parameter.

### 2.2.2. Ridge Regression

Ridge regression is a technique for reducing data overfitting by adding a small degree of bias to regression predictions. The major goal of using ridge regression is to obtain more accurate outcomes. The method allows for the estimation of coefficients in multiple regression models when a high correlation exists between the predictor variables [22]. Ridge regression may perform slightly poorly on the training set, but overall, it performs consistently well. The L2 regularization approach is used in ridge regression. The loss in ridge regression is defined as:

$$L_{ridge}(\hat{\beta}) = \sum_{i=1}^{n}(y_i - x_i'\hat{\beta})^2 + \lambda \sum_{j=1}^{m}\beta_j^2 = y - X\hat{\beta}^2 + \lambda\hat{\beta}^2 \tag{4}$$

where $x$ and $y$ are the input and output vector, respectively, $n$ is the number of samples in the training dataset, $\beta$ is the regression coefficient, and $\lambda$ is the penalty parameter.

### 2.2.3. Elastic Net (ELNET)

In ELNET, the penalty of ridge regression and LASSO, i.e., shrinkage and sparsity, are combined to reap the benefits of both ridge regression and LASSO [23]. The elastic net estimator minimizes

$$L_{enet}(\hat{\beta}) = \frac{\sum_{i=1}^{n}(y_i - x_i'\hat{\beta})^2}{2n} + \lambda\left(\frac{1-\alpha}{2}\sum_{j=1}^{m}\hat{\beta}^2 + \alpha\sum_{j=1}^{m}|\hat{\beta}_j|\right), \tag{5}$$

where $x$ and $y$ are the input and output, respectively, $n$ is the number of samples in the training dataset, $\beta$ is the regression coefficient, $\lambda$ is the penalty parameter, and $\alpha$ is the mixing parameter between ridge ($\alpha = 0$) and LASSO ($\alpha = 1$).

### 2.3. Artificial Neural Network (ANN)

An artificial neural network (ANN) is a type of non-linear machine learning technique. It has three interconnected layers, viz., input (nodes or units), hidden (one to three layers of neurons), and the output layer of neurons. Each connection is associated with a numeric number known as weight. The output hi of neuron i in the hidden layer is [24]:

$$h_i = \sigma\left(\sum_{j=1}^{N}V_{ij}x_j + T_i^{hid}\right), \tag{6}$$

Here, $\sigma$ is the activation function, $N$ is the number of input neurons, $V_{ij}$ is the weight, $x_j$ is the input to the neurons, and $T_i^{hid}$ is the threshold term of the hidden neurons.

### 2.4. Ensemble Models

Various researchers [25,26] have compared conventional machine learning models with modified and improved ensemble models. Keeping this in mind, the outputs from individual models, namely, SMLR, LASSO, ridge, ELNET, and ANN, were used as inputs, and the observed yield was used as the target variable to build the ensemble models. The ELNET model was optimized through 10-fold cross-validation with 5 repetitions using the

"caret" [27] and "glmnet" [28] packages in R. Ensemble models provide highly accurate predictions, since multiple versions of a single model are combined to arrive at a final prediction by aggregating the prediction of multiple base learners [29]. In the present study, we have used four techniques to create an ensemble model, namely, the generalized linear model (GLM), random forest (RF), cubist, and ELNET methods.

Ensemble predictors such as random forest are known to have greater accuracy, though it is difficult to understand their "black-box" prediction. In contrast, a generalized linear model (GLM) is fairly interpretable, especially when forward feature selection is employed to build the model [29]. ELNET is a type of linear regression with regularization to help prevent overfitting and built-in variable selection. On the other hand, Quinlan [30] proposed a prediction-oriented regression model known as the cubist model. The key benefit of the cubist method is the addition of multiple training committees to balance case weights.

### 2.5. Evaluation of Model Performance

The performance of the models was evaluated using the $R^2$, the root mean square error (RMSE), the normalized root mean square error (nRMSE), the mean biased error (MBE), the mean absolute error (MAE) and the concordance correlation coefficient ($\rho_c$). A value of $R^2$ and $\rho_c$ close to 1 and of RMSE, MBE, and MAE near to 0 indicates better model performance. The positive and negative values of the MBE indicate over- and underestimation, respectively. In addition to this, the model performance is considered as excellent, good, fair, or poor based on the value of the nRMSE lying between 0–10%, 10–20%, 20–30%, or >30%, respectively.

The formulas of the model evaluation measures are shown below:

$$RMSE = \sqrt{\frac{\sum_{i=1}^{n}(y_i - \hat{y}_i)^2}{n}}, \tag{7}$$

$$nRMSE = \sqrt{\frac{\sum_{i=1}^{n}(y_i - \hat{y}_i)^2}{n}} \times \frac{100}{\bar{A}}, \tag{8}$$

$$R^2 = \left(\frac{\frac{1}{n}\sum_{i=1}^{n}\left(y_i - \bar{y}\right)\left(\hat{y}_i - \bar{\hat{y}}\right)}{\sigma_y \sigma_{\hat{y}}}\right)^2, \tag{9}$$

$$MBE = \frac{1}{n}\sum_{i=1}^{n}(y_i - \hat{y}_i) \tag{10}$$

$$MAE = \frac{\sum_{i=1}^{n}|y_i - \hat{y}_i|}{n}, \tag{11}$$

$$\rho_c = \frac{2\rho\sigma_x\sigma_y}{\sigma_x^2 + \sigma_y^2 + (\mu_x - \mu_y)^2}, \tag{12}$$

Here, $y_i$ is the observed value, and $\hat{y}_i$ is the predicted value for i=1, 2, . . . . . . , n. $\sigma_y$ and $\sigma_{\hat{y}}$ are the standard deviation of actual and predicted observations, respectively. $\bar{y}$ and $\bar{\hat{y}}$ denote the average of the observed and predicted values, $\mu_x$ and $\mu_y$ are the means for the observed and predicted values, and $\rho$ is the correlation coefficient between the observed and predicted values.

## 3. Results

### 3.1. Evaluation of the Model Performance

The results pertaining to all models' performances and equations developed using these models are shown in Tables 2 and 3, respectively. The weighted indices, which were

calculated based on the correlation of the weather variables with the dependent variable, have more importance, and it can be seen here that the inclusion frequency of the weighted weather indices was much higher than that of the simple weather indices in all the models for all three locations. This fact can be seen in the developed equations (Table 3), which signifies the impact of the weighted weather indices.

**Table 2.** Comparison of different models for rice yield in training and testing datasets.

|  |  | Criteria/Model | SMLR [1] | LASSO [2] | Ridge [3] | ELNET [4] | ANN [5] |
|---|---|---|---|---|---|---|---|
| Raipur | Calibration | $R^2_{cal}$ | 0.89 | 0.974 | 0.715 | 0.99 | 1.00 |
|  |  | $RMSE_{cal}$ | 0.142 | 0.069 | 0.25 | 0.041 | 0.002 |
|  |  | $nRMSE_{cal}$ | 10.62 | 5.191 | 18.741 | 3.089 | 0.155 |
|  |  | $MBE_{cal}$ | 0.757 | 0.00 | 0.00 | 0.00 | 0.00 |
|  |  | $MAE_{cal}$ | 0.75 | 0.05 | 0.18 | 0.03 | 0.001 |
|  | Validation | $R^2_{val}$ | 0.16 | 0.395 | 0.801 | 0.46 | 1.00 |
|  |  | $RMSE_{val}$ | 0.647 | 0.632 | 1.019 | 0.633 | 0.003 |
|  |  | $nRMSE_{val}$ | 28.368 | 27.709 | 44.633 | 27.748 | 0.167 |
|  |  | $MBE_{val}$ | −0.063 | −0.393 | −0.906 | −0.417 | −0.0019 |
|  |  | $MAE_{val}$ | 0.28 | 0.39 | 0.91 | 0.43 | 0.003 |
| Surguja | Calibration | $R^2_{cal}$ | 0.89 | 0.938 | 0.732 | 0.92 | 1.00 |
|  |  | $RMSE_{cal}$ | 0.113 | 0.089 | 0.21 | 0.103 | 0.004 |
|  |  | $nRMSE_{cal}$ | 9.86 | 7.787 | 18.501 | 9.023 | 0.326 |
|  |  | $MBE_{cal}$ | −0.017 | 0.00 | 0.00 | 0.00 | −0.0007 |
|  |  | $MAE_{cal}$ | 0.09 | 0.07 | 0.18 | 0.08 | 0.003 |
|  | Validation | $R^2_{val}$ | 0.07 | 0.158 | 0.007 | 0.002 | 0.99 |
|  |  | $RMSE_{val}$ | 0.206 | 0.235 | 0.292 | 0.191 | 0.004 |
|  |  | $nRMSE_{val}$ | 14.104 | 16.079 | 19.778 | 13.072 | 0.318 |
|  |  | $MBE_{val}$ | 0.135 | 0.193 | −0.246 | 0.100 | 0.002 |
|  |  | $MAE_{val}$ | 0.19 | 0.19 | 0.25 | 0.17 | 0.004 |
| Bastar | Calibration | $R^2_{cal}$ | 0.70 | 0.93 | 0.769 | 0.90 | 0.74 |
|  |  | $RMSE_{cal}$ | 0.227 | 0.096 | 0.21 | 0.127 | 0.214 |
|  |  | $nRMSE_{cal}$ | 19.79 | 8.385 | 19.09 | 11.09 | 18.568 |
|  |  | $MBE_{cal}$ | −0.126 | 0.00 | 0.00 | 0.00 | 0.033 |
|  |  | $MAE_{cal}$ | 0.19 | 0.08 | 0.19 | 0.11 | 0.19 |
|  | Validation | $R^2_{val}$ | 0.31 | 0.568 | 0.242 | 0.48 | 0.0009 |
|  |  | $RMSE_{val}$ | 0.435 | 0.322 | 0.294 | 0.255 | 0.199 |
|  |  | $nRMSE_{val}$ | 29.764 | 22.043 | 20.130 | 17.444 | 13.645 |
|  |  | $MBE_{val}$ | −0.381 | 0.158 | −0.244 | 0.053 | −0.143 |
|  |  | $MAE_{val}$ | 0.38 | 0.26 | 0.25 | 0.21 | 0.17 |

[1] SMLR = stepwise multiple linear regression, [2] LASSO = least absolute shrinkage and selection operator, [3] Ridge = ridge regression, [4] ELNET = elastic net, [5] ANN = artificial neural network.

### 3.1.1. Stepwise Multiple Linear Regression (SMLR)

The performance of SMLR was good at the calibration stage for all the three districts, with a $R^2$ value of 0.89 for Raipur and Surguja and 0.70 for Bastar with a RMSE of 0.142, 0.113, and 0.227 t/ha respectively. The nRMSE values for the calibration stage were also less than 20%, indicating good model performance, though at the validation stage, the

performance of SMLR was poor for all three districts, with $R^2$ values of 0.16, 0.07, and 0.31 for Raipur, Surguja, and Bastar, respectively. In addition to this, the nRMSE value at the validation stage indicated good model performance for Surguja (nRMSE = 14.10%) and fair performances for Raipur (nRMSE = 28.37%) and Bastar (nRMSE = 29.76%). The MBE value indicates that the model underestimated the crop yield for the Raipur (MBE = −0.063 t/ha) and Bastar (MBE = −0.381t/ha) districts, and for the Surguja (MBE = 0.135 t/ha) district, it was slightly overestimated at the validation stage. The MAE values also suggest better model performance for Surguja and Bastar as compared to the Raipur district.

**Table 3.** Equations developed by different models for rice yield forecasting.

| Model | District | Equation |
|---|---|---|
| SMLR [1] | Raipur | Yield = 1.562 + 0.06 × Time + 0.043 × Z61 + 0.001 × Z361 |
| | Surguja | Yield = 0.46 + 0.04 × Time + 0.0000908 × Z131 |
| | Bastar | Yield = 1.16 + 0.005 × Z261 − 0.001 × Z361 + 0.000037 × Z240 |
| LASSO [2] | Raipur | Yield = 1.339 + 0.274 × Time + 0.025 × Z20 + 0.107 × Z41 + 0.038 × Z120 − 0.027 × Z140 + 0.063 × Z141 + 0.002 × Z230 + 0.063 × Z361 − 0.058 × Z560 + 0.051 × Z561 |
| | Surguja | Yield = 1.15 + 0.214 × Time + 0.087 × Z131 + 0.025 × Z151 − 0.005 × Z160 + 0.005 × Z340 + 0.039 × Z351 + 0.013 × Z451 + 0.05 × Z461 − 0.027 × Z560 |
| | Bastar | Yield = 1.15 + 0.193 × Time + 0.007 × Z11 + 0.001 × Z40 + 0.072 × Z41 − 0.175 × Z160 − 0.021 × Z360 |
| Ridge Regression | Raipur | Yield = 1.339 + 0.027 × Time − 0.006 × Z10 + 0.011 × Z11 + 0.006 × Z20 + 0.001 × Z21 + 0.013 × Z30 + 0.006 × Z31 + 0.005 × Z40 + 0.013 × Z41 − 0.004 × Z50 + 0.001 × Z51 − 0.008 × Z60 + 0.013 × Z61 + 0.005 × Z120 + 0.006 × Z121 + 0.015 × Z130 + 0.006 × Z131 − 0.001 × Z140 + 0.01 × Z141 − 0.004 × Z150 − 0.002 × Z151 − 0.008 × Z160 + 0.013 × Z161 + 0.015 × Z230 + 0.006 × Z231 + 0.011 × Z240 + 0.004 × Z241 + 0.002 × Z250 + 0.002 × Z251 − 0.007 × Z260 + 0.013 × Z261 + 0.014 × Z340 + 0.006 × Z341 + 0.010 × Z360 + 0.002 × Z361 − 0.001 × Z450 + 0.002 × Z451 − 0.008 × Z460 + 0.013 × Z461 − 0.012 × Z560 + 0.009 × Z561 |
| | Surguja | Yield = 1.150 + 0.023 × Time − 0.002 × Z10 + 0.005 × Z11 − 0.007 × Z20 − 0.003 × Z21 − 0.003 × Z30 + 0.012 × Z31 + 0.015 × Z40 + 0.014 × Z41 + 0.002 × Z50 + 0.007 × Z51 − 0.007 × Z60 + 0.009 × Z61 − 0.005 × Z120 + 0.007 × Z121 − 0.003 × Z130 + 0.013 × Z131 + 0.009 × Z140 − 0.002 × Z141 − 0.001 × Z150 + 0.007 × Z151 − 0.006 × Z160 + 0.009 × Z161 − 0.005 × Z230 + 0.011 × Z231 − 0.002 × Z240 − 0.001 × Z241 − 0.006 × Z250 − 0.008 × Z260 + 0.011 × Z261 − 0.002 × Z340 + 0.012 × Z341 − 0.004 × Z360 + 0.006 × Z361 + 0.006 × Z450 + 0.009 × Z451 − 0.004 × Z460 + 0.008 × Z461 −0.007 × Z560 + 0.010 × Z561 |
| | Bastar | Yield = 1.150 + 0.020 ×Time − 0.003 × Z10 + 0.008 × Z11 − 0.001 × Z20 −0.001 × Z21 + 0.004 × Z30 + 0.002 × Z31+ 0.006 × Z40 + 0.006 × Z41 − 0.002 × Z50 − 0.001 × Z51 − 0.014 × Z60 + 0.014 × Z61 − 0.001 × Z120 +0.002 × Z121 + 0.003 × Z130 + 0.001 × Z131 + 0.006 × Z140 + 0.008 × Z141 − 0.004 × Z150 − 0.002 × Z151 − 0.016 × Z160 + 0.014 × Z161 + 0.004 × Z230 + 0.002 × Z231 + 0.005 × Z240 + 0.003 × Z241 − 0.003 × Z250 − 0.002 × Z251 − 0.017 × Z260 + 0.017 × Z261 + 0.004 × Z340 + 0.002 × Z341 + 0.003 × Z350 +0.001 × Z351 − 0.008 × Z360 − 0.003 × Z361 + 0.001 × Z450 + 0.001 × Z451 − 0.014 × Z460 + 0.015 × Z461 − 0.014 × Z560 + 0.011 × Z561 |
| ELNET [3] | Raipur | Yield = −6.069 + 0.049 × Time + 0.004 × Z20 −0.001 × Z40 + 0.045 × Z41 |
| | Surguja | Yield = 0.044 + 0.036 × Time + 0.002 × Z41 |
| | Bastar | Yield = 1.547 + 0.028 × Time |

[1] SMLR = stepwise multiple linear regression, [2] LASSO = least absolute shrinkage and selection operator, [3] ELNET = elastic net. All the other abbreviations are available in Table 1.

The use of both weighted and unweighted weather indices could lead to a strong multicollinearity problem. All the models except SMLR can deal with the multicollinearity problem. The three models, viz., LASSO, ELNET and ridge regression, are designed in such a way that they can deal with the multicollinearity problem [31,32]. In ANN too, multicollinearity might be not a problem, as it uses non-linear activation functions. Collinearity would be an issue when there is no regularization scheme and non-linearity. Hence, to check the multicollinearity problem for SMLR, the variance inflation factor (VIF) was calculated for all three locations. The results are included in Table 4. The results revealed no or moderate multicollinearity, with VIF values < 5 [33].

**Table 4.** Rice yield prediction models using SMLR.

| Districts | Predictor Variables | Coefficient | VIF | $R^2_{val}$ | $RMSE_{val}$ |
|---|---|---|---|---|---|
| Raipur | Constant | 1.562 | | 0.75 | 0.647 |
| | Time | 0.06 | 1.262 | | |
| | Z61 | 0.043 | 1.595 | | |
| | Z361 | 0.001 | 1.583 | | |
| Surguja | Constant | 0.46 | | 0.07 | 0.206 |
| | Time | 0.04 | 1.035 | | |
| | Z131 | 0.00009 | 1.035 | | |
| Bastar | Constant | 1.16 | | 0.31 | 0.435 |
| | Z261 | 0.005 | 1.390 | | |
| | Z361 | −0.001 | 1.581 | | |
| | Z240 | 0.000037 | 1.184 | | |

### 3.1.2. Least Absolute Shrinkage and Selection Operator (LASSO)

The results of the LASSO analysis revealed that the value of $R^2$ ranged from 0.93 to 0.97. The highest $R^2$ was observed for the Raipur district ($R^2$ = 0.97), with a RMSE value of 0.069 t/ha, followed by Surguja ($R^2$ = 0.94) and Bastar ($R^2$ = 0.93), with RMSE values of 0.089 t/ha and 0.096 t/ha, respectively. On the other hand, in the validation stage, the $R^2$ value ranged from 0.158 to 0.568. The highest $R^2$ was observed for Bastar ($R^2$ = 0.57), with RMSE 0.322 t/ha, whereas the lowest $R^2$ was observed for Surguja ($R^2$ = 0.158), with RMSE 0.235 t/ha. Moreover, at the validation stage, the performance of the LASSO model was good for Surguja (nRMSE = 16.08%) and fair for Raipur (nRMSE = 27.71%) and Bastar (nRMSE = 22.04%). Hence, the LASSO model can be used to forecast the rice yield for the Bastar district. Despite good model performance at the calibration stage, the MBE value indicated underestimation of the crop yield for the Raipur (MBE = −0.393) district and overestimation for Surguja (MBE = 0.193) and Bastar (MBE = 0.158) at the validation stage. The variable importance developed using LASSO is shown in Figure 3. During the calibration stage, MAE values were near to zero for all three locations (0.05, 0.07, and 0.08), suggesting excellent model performance, and during the validation stage, MAE values were found to be 0.39, 0.19, and 0.26 for Raipur, Surguja, and Bastar, respectively.

### 3.1.3. Ridge Regression

For ridge regression, at the calibration stage, the maximum $R^2$ value was found for the Bastar (0.769) district, with a RMSE of 0.21 t/ha, and the minimum $R^2$ was recorded for Raipur (0.715), with a RMSE of 0.25 t/ha. In addition to this, the value of the nRMSE showed good model performance (nRMSE < 20%) for all the districts. The RMSE statistic for the validations showed that the performance of the ridge regression model was good for Surguja (nRMSE = 19.978%), fair for Bastar (nRMSE = 20.130%), and poor for Raipur, (nRMSE = 44.633%). Contrary to this, at the validation stage, the value of $R^2$ was good for Raipur ($R^2$ = 0.80) and poor for Surguja ($R^2$ = 0.01) and Bastar ($R^2$ = 0.24). The MBE value indicated underestimation of the crop yield for all three districts of Chhattisgarh at the validation stage. The MAE values were found to be 0.18 for both Raipur and Surguja, and 0.19 for the Bastar district at the calibration stage. During validation, the MAE was found to

be 0.91 for Raipur and 0.25 for both Surguja and Bastar, suggesting poor model performance. Therefore, overall, ridge regression is not a good choice to provide the rice yield forecast for all three studied districts of Chhattisgarh. The variable importance developed using ridge regression is shown in Figure 4.

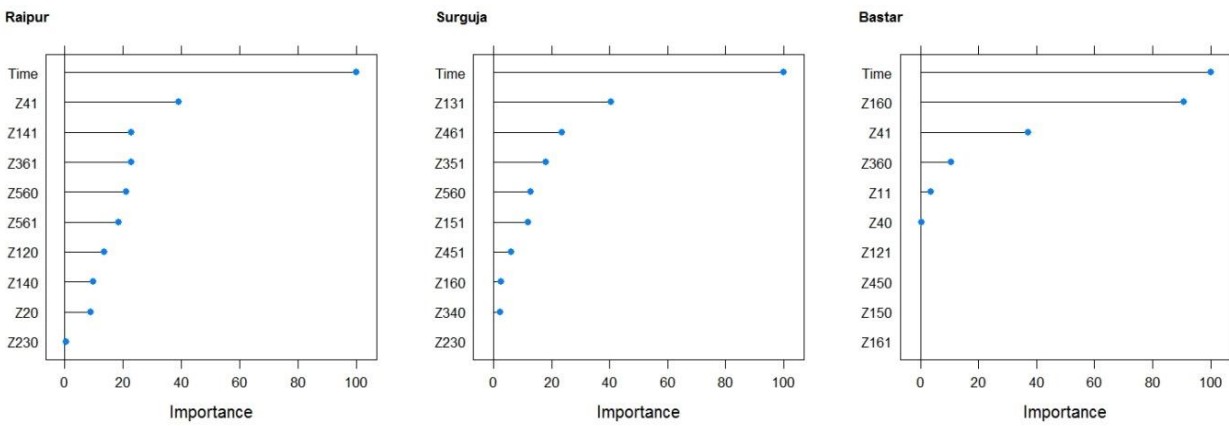

**Figure 3.** Variable importance of LASSO for all three locations.

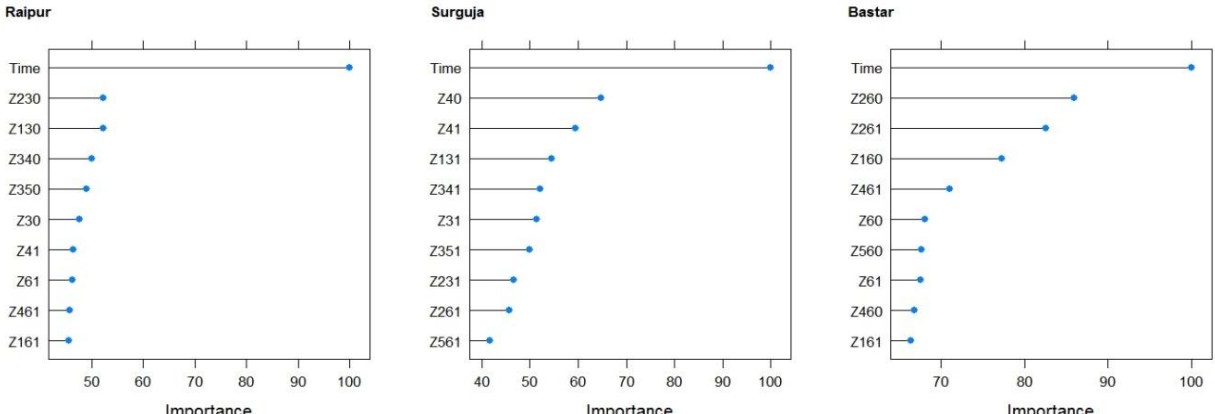

**Figure 4.** Variable importance of ridge regression for all three locations.

### 3.1.4. Elastic Net (ELNET)

For the ELNET model, the values of the $R^2$, RMSE, nRMSE, and MBE were excellent for all the districts considered in the study. The value of $R^2$ was 0.99, 0.92, and 0.90 for the Raipur, Surguja, and Bastar districts, respectively, at the calibration stage. The RMSE of calibrated data ranged from 0.041 to 0.127 t/ha. During the validation stage, the value of $R^2$ was good for Raipur ($R^2$ = 0.46, nRMSE = 27.75%) and Bastar ($R^2$ = 0.48, nRMSE = 17.44%), whereas it was poor for Surguja ($R^2$ = 0.002, nRMSE = 13.07%). The MBE values also indicated underestimation for the Raipur (MBE = $-0.417$ t/ha) district and overestimation for the Surguja (MBE = 0.100 t/ha) and Bastar (MBE = 0.053 t/ha) districts. Excellent MAE values were noticed during the calibration stage of the ELNET model, i.e., 0.03 for Raipur, 0.08 for Surguja, and 0.11 for Bastar. During the validation stage, the values of the MAE were 0.43, 0.17, and 0.21 for Raipur, Surguja, and Bastar, respectively. The overall findings concluded that the ELNET model can be used for the rice yield forecasting of Bastar. The variable importance developed using ELNET is shown in Figure 5.

### 3.1.5. Artificial Neural Network (ANN)

The results of the analysis showed that the performance of the artificial neural network (ANN) was excellent for the Raipur and Surguja districts, with a $R^2$ value of 1.00 and a RMSE of 0.002 and 0.004 t/ha, respectively, during calibration. Meanwhile, during validation, the $R^2$ was 1.00 and 0.99 with a RMSE 0.003 and 0.004 t/ha, respectively, for

these locations. In addition to this, the value of the nRMSE for the Raipur and Surguja districts was less than 10% for the calibration as well as the validation stage, indicating excellent model performance. Although for Bastar, the model performance was good at the calibration stage ($R^2$ = 0.74, nRMSE = 18.57%), during validation, the model gave a poor performance ($R^2$ = 0.00). The MBE values for Raipur and Surguja were close to zero. For Bastar (MBE = −0.143 t/ha), an underestimation of the rice yield at the validation stage was indicated. The MAE values also indicated the best ANN model performance was for the Raipur district, with 0.001 and 0.003 for the calibration and validation stages, respectively. Comparatively higher values of the MAE were noticed for the Bastar district, i.e., 0.19 and 0.17, for the calibration and validation stages, respectively. Therefore, for Raipur and Surguja, the ANN is a good choice for rice yield forecasting. The weights and biases for the developed ANN models can be found in Tables S1–S3 for Raipur, Surguja and Bastar respectively. Structure of neural network for developed ANN models illustrated in Figure S1 for Raipur, Figure S2 for Surguja and Figure S3 for Bastar. The variable importance developed using the ANN is shown in Figure 6.

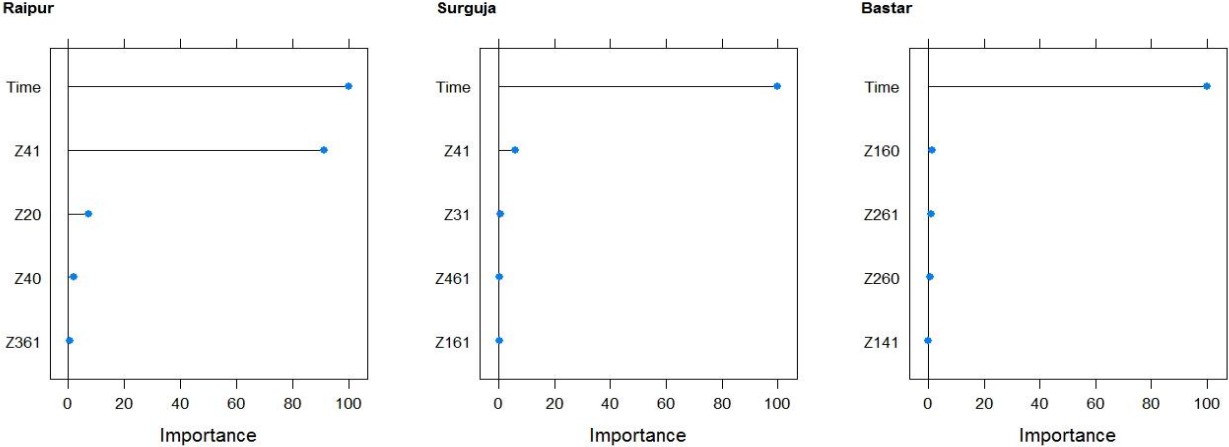

**Figure 5.** Variable importance of ELNET for all three locations.

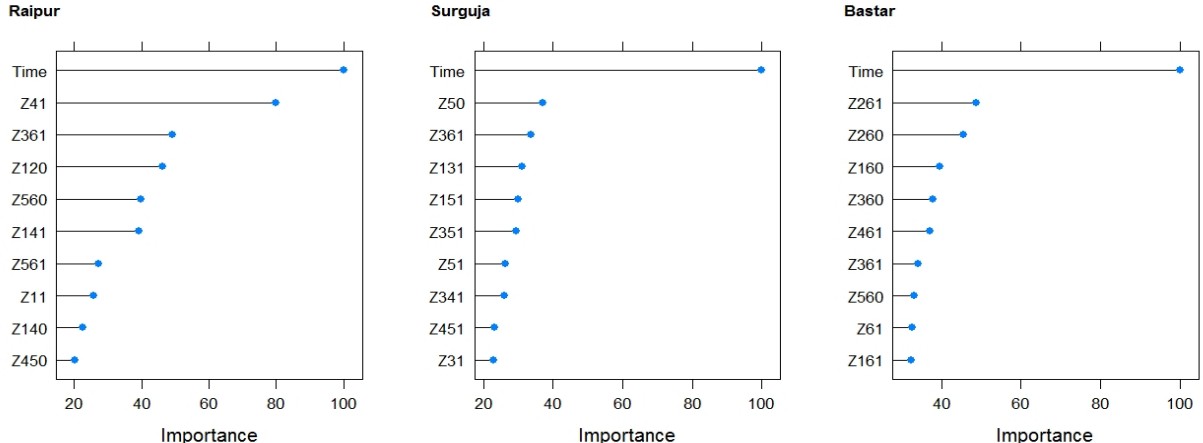

**Figure 6.** Variable importance of ANN for all three locations.

Another comparison was made between the developed models to check the importance of time as a factor. Hence, we ran all the models for each location by (1) using time as the only independent factor and (2) using only weather variables as independent variables. It the first case, only two models could be developed, viz., SMLR and ANN. The other three models (LASSO, ELENT, and ridge regression) need more than one independent variable to predict yield. Hence, in using time only, these three models could not be developed. The results of the two cases and improvements are provided in the Table 5. In the first

case, both the SMLR and ANN models showed poor performance based on the $R^2$ value. On the other hand, when time was excluded, the performances of the models during the calibration stage were found to be either similar or decreased based on the $R^2$ value.

**Table 5.** Comparison of models based on $R^2$ value for calibration stage.

| Location | Model | Only Time as Factor | Time Excluded as a Factor | Both Time and Weather Variables Included | Improvement in Time and Weather Variables |
|---|---|---|---|---|---|
| Raipur | SMLR | 0.57 | 0.68 | 0.89 | 30.88% |
| | LASSO | - | 0.67 | 0.97 | 44.78% |
| | Ridge | - | 0.60 | 0.71 | 18.33% |
| | ELNET | - | 0.65 | 0.99 | 52.31% |
| | ANN | 0.57 | 0.78 | 1.00 | 28.21% |
| Surguja | SMLR | 0.50 | 0.69 | 0.89 | 28.99% |
| | LASSO | - | 0.69 | 0.94 | 36.23% |
| | Ridge | - | 0.66 | 0.73 | 10.61% |
| | ELNET | - | 0.66 | 0.92 | 39.39% |
| | ANN | 0.50 | 0.57 | 1.00 | 75.44% |
| Bastar | SMLR | 0.29 | 0.71 | 0.70 | −1.41% |
| | LASSO | - | 0.75 | 0.93 | 24.00% |
| | Ridge | - | 0.71 | 0.77 | 8.45% |
| | ELNET | - | 0.75 | 0.90 | 20.00% |
| | ANN | 0.39 | 0.66 | 0.74 | 12.12% |

### 3.2. Ensemble Models

In order to improve the accuracy of predictive analytics, ensemble modeling involves running two or more separate but related analytical models, synthesizing their results, and then aggregating them. In the present study, we used the GLM, RF, cubist, and ELNET approaches to form an ensemble model by using the output of individual models, i.e., SMLR, LASSO, ANN, ELNET, and ridge regression. The results of the ensemble model are demonstrated in Table 6. The findings show that for all the locations, the performance of the ensemble model was better as compared to the individual models. For Raipur and Surguja, the performances of all the ensemble methods was comparable, whereas for Bastar, random forest (RF) performed better as compared to the GLM, cubist, and ELNET approaches.

**Table 6.** Comparison of different ensemble models for rice yield in training and testing datasets.

| Ensemble Methods | Raipur Calibration | | | Raipur Validation | | |
|---|---|---|---|---|---|---|
| | $R^2_{cal}$ | $\rho_{cal}$ | $RMSE_{cal}$ | $R^2_{val}$ | $\rho_{val}$ | $RMSE_{val}$ |
| GLM [1] | 0.999 | 0.941 | 0.002 | 0.999 | 0.749 | 0.001 |
| RF [2] | 0.926 | 0.890 | 0.183 | 0.955 | 0.731 | 0.107 |
| Cubist | 0.999 | 0.941 | 0.002 | 0.999 | 0.749 | 0.001 |
| ELNET | 0.999 | 0.941 | 0.018 | 0.999 | 0.749 | 0.014 |
| | Surguja calibration | | | Surguja validation | | |
| GLM [1] | 0.999 | 0.941 | 0.002 | 0.999 | 0.749 | 0.003 |
| RF [2] | 0.978 | 0.927 | 0.055 | 0.906 | 0.703 | 0.128 |
| Cubist | 0.999 | 0.749 | 0.003 | 0.999 | 0.749 | 0.005 |
| ELNET | 0.999 | 0.941 | 0.01 | 0.999 | 0.749 | 0.014 |
| | Bastar calibration | | | Bastar validation | | |
| GLM [1] | 0.561 | 0.676 | 0.262 | 0.529 | 0.516 | 0.364 |
| RF [2] | 0.855 | 0.839 | 0.163 | 0.806 | 0.641 | 0.222 |
| Cubist | 0.523 | 0.634 | 0.274 | 0.809 | 0.651 | 0.202 |
| ELNET | 0.521 | 0.609 | 0.276 | 0.823 | 0.635 | 0.205 |

[1] GLM = general linear model, [2] RF = random forest.

## 4. Discussion

### 4.1. Influence of Weather Parameters on Rice Crop

There is a profound influence of weather parameters on rice yield, and there is extensive literature available on the effect of temperature on rice yield [34–36]. The mean weekly temperature for the study regions during the rice-growing season varied from 23.7 to 31.0 °C, 21.2 to 29.6 °C, and 21.2 to 28.9 °C for Raipur, Surguja, and Bastar, respectively, which is quite close to the ideal temperature needed for rice growth i.e., 15–18 to 30–33 °C [37]. However, the maximum temperature sometimes exceeded 35 °C, and these extreme temperatures have a negative impact on rice yield and growth [38]. The rate of photosynthesis, respiration, spikelet sterility, and length of the growing season are all impacted by temperature, which also has an impact on crop yield [39,40]. Higher temperatures have been observed to shorten the grain filling time, which reduces crop yield as well as the grain quality. Moreover, rice yield is positively influenced by solar radiation, as it directly affects biomass accumulation [40]. Reduced sun exposure, especially during the reproductive and ripening stages, results in a reduction in the crop yield [41].

Temperature, relative humidity, and solar radiation were the major weather parameters with the maximum impacts on the rice yield. Solar radiation was selected to be an important variable for rice yield using the ANN and LASSO models for the Raipur and Bastar districts, respectively, which is in agreement with previous studies [42], and the maximum temperature and relative humidity was found to be the most significant predictors of rice yield for the Surguja district.

### 4.2. Model Cross-Comparison

In the face of climate change, population growth, and food demand, timely, accurate, and reliable crop yield estimation is much more important than ever before in crop management, food security assessment, and policymaking. In the present study, an attempt has been made to discover the best prediction model for rice yield forecasting for three districts of different agro-climatic zones of Chhattisgarh, India. By combining the weather data with publicly available agricultural production data, a comparison was made among the models developed using multiple approaches, including SMLR, ANN, LASSO, ELNET, and ridge regression. Different statistical measures were used to capture the performance of each model.

The finding showed that based on the value of the coefficient of determination ($R^2$), SMLR performed well at calibration, whereas at validation, it performed poorly for all the districts, which might be due to the overfitting of data at the calibration stage. On the other hand, the performance of LASSO was good at calibration for all three districts, whereas it was poor at validation for Raipur and Surguja. The LASSO technique was found to be the most accurate for rice yield forecasting for the Bastar district at both the calibration and validation stages. This finding is consistent with the findings of Kumar et al. [43], Singh et al. [44], and Parul et al. [45], who observed that LASSO performed better than SMLR to some extent. The performance of LASSO was good, as the model used the shrinkage technique to deal with the issue of multicollinearity, thus keeping only the most significant variable in the study.

Moreover, for ridge regression, the performance of the model was good at the calibration stage, whereas it was poor at the validation stage for all three districts based on the value of the $R^2$ and nRMSE. For the ELNET model, at the calibration stage, the value of the $R^2$, RMSE, nRMSE, and MBE was excellent for all the districts considered in the study. However, during the validation stage, model performance was good for Raipur and Bastar and poor for Surguja. The overall findings concluded that the ELNET model can be used for the rice yield forecasting for Bastar. The good performance of ELNET may be due to the inclusion of Lasso and ridge penalties, so the model hypothesis space is much broader with ELNET.

Likewise, the ANN works in a non-linear fashion, having a potential advantage in the analysis of variables with complex correlations compared to regression models. The

performance of the ANN was excellent for the Raipur and Surguja districts during the calibration as well as the validation stage. Meanwhile, for Bastar, the model performance was good at the calibration and poor at validation stage. Therefore, for Raipur and Surguja, the ANN is a good choice for rice yield forecasting. The results supported the study by Aravind et al. [16], which demonstrated that the ANN performed better than SMLR, LASSO, and ELNET in the Patiala district of Punjab.

It is worth noticing that different weighted and unweighted climatic indices were selected in different models and districts. All the three districts of the state are part of three different climatic zones of Chhattisgarh state (India), and their climatic conditions are very different from each other. Surguja belongs to the northern hilly region, Raipur is part of the central plains, and Bastar is part of the southern plateau. Hence, it is understandable that different climatic variables could affect the rice yield in different regions.

### 4.3. Limitations

Many times, commonly used linear regression models to predict crop yields such as SMLR, LASSO, ridge, and ELNET fail to fit complex nonlinear relationships between crop yield and weather data. The poor performance of ridge regression compared to LASSO and ELNET may be due to the inclusion of all predictor variables in the final model. In the case of multiple highly collinear variables, LASSO regression randomly selects one of them, which can lead to incorrect interpretation. The alternative to these mentioned approaches is an ANN, but the ANN is not perfect when it comes to learning patterns, since crop yield is affected by a variety of factors, which are unpredictable and inconsistent. Additionally, we used a single hidden layer for ANN fitting. The inclusion of multiple hidden layers with a large amount of training data may further improve the predictability of the ANN model. Furthermore, an attempt was also made to form an ensemble model. The performance of the ensemble model may not be better than the best individual model, as the ensemble model is built using the individual models only. However, it will definitely be better than the individual model having the worst performance. The biggest limitation of the present study was the unavailability of long-term rice yield and weather data.

### 5. Conclusions

In the present study, five different methods, viz., SMLR, LASSO, ELNET, ridge regression, and an ANN, were used to study the relationship of yield to weather parameters for three districts of Chhattisgarh, India. The overall ranking based on the RMSE and nRMSE values during validation revealed that the ANN performed the best for the Raipur ($R^2 = 1$) and Surguja ($R^2 = 1$ and 0.99) districts as compared to other models, and for the Bastar district, LASSO ($R^2 = 0.93$ and 0.57) and ELNET ($R^2 = 0.90$ and 0.48) performed better compared to other models included in the study. The study also showed that the performance of SMLR, LASSO, ELNET, and ridge regression was good during calibration but not during the process of validation, which may be due to overfitting. In the ensemble model, the performance was found to be better compared to the individual models. For Raipur and Surguja, the performance of all the ensemble methods was excellent and comparable, whereas for Bastar, random forest (RF) performed better, with $R^2 = 0.85$ and 0.81 for calibration and validation, respectively, as compared to the GLM, cubist, and ELNET approaches.

### Future Research

There is no doubt that deep learning has rapidly expanded in the field of agriculture since 2019, including the prediction of crop yields. In future studies, focus should be placed on some deep learning techniques, such as CNN, DNN, RNN, LSTM, MLP, R-CNN, and faster R-CNN, both individually as well as in different combinations.

**Supplementary Materials:** The following supporting information can be downloaded at: https://www.mdpi.com/article/10.3390/su15032786/s1, Table S1: Weight and biases of ANN model developed for Raipur district, Table S2: Weight and biases of ANN model developed for Surguja district, Table S3: Weight and biases of ANN model developed for Bastar district, Figure S1: Structure of neural network for rice yield prediction of Raipur, Figure S2: Structure of neural network for rice yield prediction of Surguja, Figure S3: Structure of neural network for rice yield prediction of Bastar.

**Author Contributions:** Conceptualization, A.S. and P.S.; methodology, P.S. and B.D.; software, P.S. and B.D.; formal analysis, A.S. and P.S.; writing—original draft preparation, A.S., P.S., A.S.N. and P.K.J.; writing—review and editing, S.S. (Surendra Singh) and S.S. (Shikha Singh).; visualization, A.S., P.K.J., S.S. (Surendra Singh) and S.S. (Shikha Singh). All authors have read and agreed to the published version of the manuscript.

**Funding:** This research received no external funding.

**Institutional Review Board Statement:** The research of our manuscript was privately funded and received no external funding. All subjects gave their informed consent for inclusion before they participated in the study in accordance with the Declaration of Helsinki.

**Informed Consent Statement:** Not applicable.

**Data Availability Statement:** The dataset used and/or analyzed during the current study is available from the corresponding authors on reasonable request.

**Acknowledgments:** The authors would like to thank the anonymous reviewers for their valuable comments that helped shape the key messages of the manuscript.

**Conflicts of Interest:** The authors declare no conflict of interest.

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
