# Peer review of "Comparative Analysis of Statistical and Machine Learning Techniques for Rice Yield Forecasting for Chhattisgarh, India"

_sustainability, doi:10.3390/su15032786_

Round 1

Reviewer 1 Report

The study deals with “Comparative Analysis of Statistical and Machine Learning Techniques for Rice Yield Forecasting for Chhattisgarh, India. Overall a good effort has been made by the authors to write the manuscript. But the following comments should be addressed before it can be accepted.
1) Why the authors have particularly chosen the described ML models for the prediction of Rice Yield?
2) The literature review about the developed ML models must be improved by incorporating its latest application in the field of engineering (e.g., https://link.springer.com/article/10.1007/s00366-021-01329-3; https://www.sciencedirect.com/science/article/abs/pii/S1568494621005160; and https://www.icevirtuallibrary.com/doi/abs/10.1680/jgeen.19.00297)
3) Besides R2, MBE and RMSE, add more indices to stress upon the accuracy of the developed ELM models. Good example for indices can be found in “Predicting the settlement of geosynthetic-reinforced soil foundations using evolutionary artificial intelligence technique”.
4) Kindly provide the weights and biases of the developed ANN models within manuscript. This is important as without it, it would not be possible to replicate the results for the future researchers.
5) Provide the architecture of optimum neural network.
6) Kindly unify the style of Fig. 4 and 5.
7) Add some more quantitative analysis for improving the quality of the manuscript. See “Multivariate adaptive regression splines model for reinforced soil foundations”.

Author Response

Please find the response to reviewer attached. All the changes are mentioned as line number in the revised manuscript with track changes.

Reviewer 2 Report

The paper presents an empirical study for estimating the rice yield with the use of weather parameters included models.

All of the used models are well known by the scientific community. More importantly, the yield of a crop should not be considered as affected by only climatic parameters. There are several biophyscial factors and farming practice affects on the yield (irrigation, fertilization, planting date etc)

Models just work in a correlative way, no spatial component was included.

In the abstract and Methodology, some of the used models/algorithms are not mentioned (such as RF)

The introduction is very limited and does not credit the recent papers well. Also, it should clearly present what is the novelty and significance of the work.

Limitations should not be in conclusions,

Both Abstract and conclusion should provide a quantitative summary of the findings.

Author Response

Please find the response to reviewer attached. All the changes are highlighted as line number in the track changed version of the revised manuscript. 

Thanks,

Authors

Reviewer 3 Report

In this paper, five methods were used and compared, including statistical and machine learning approaches in studying the relationship between yield and weather parameters in three districts of Chhattisgarh, India. Below are a few comments:

1.     Please explain why weighted weather indices are necessary or helpful. Also, please show more details about how your weighting approach and justify the solidity.

2.     Please explain more clearly for equation 1 and 2. What do Z, j and prime sign mean?

3.     Please explain why using both weighted and unweighted indices together. This could lead to strong multicollinearity problem.

4.     Please make Table 3 clearer and neater.  

5.     Please explain why different climate variables were selected in different models and districts.

6.     It seems that Time is always the most important factor in both statistical and ML models, and much more important than climate factors. However, my understanding about this study is that the authors are trying to make use of climate factors for rice yield simulation and forecast, but the authors didn’t explain why they include Time as a factor. Given that Time is so influential, I suggest the authors do a comparison between two sets of models: 1. Use Time as the only factor in your models; 2. Remove Time and only use climate variables in your models. Please compare the model performance of the two sets of models using your metrics, which would help us to understand if Time is truly dominant in facilitating accurate yield forecasts or not.

Author Response

(The authors gave the same response as above.)

Round 2

Reviewer 1 Report

I believe the authors have made good effort to improve the quality of the manuscript. And can be accepted after a few minor revisions.

Please carefully go through it to improve the overall structure and language of the paper.

Some references mentioned in the manuscript/revision can't be found in the reference list. e.g., "Predicting the settlement of geosynthetic-reinforced settlement using evolutionary artifical intelligence ". Also check for others. Kindly cite adequately both in text and in the reference list. 

Author Response

Please check our response in the letter attached.

Thanks,

Authors

Reviewer 2 Report

Authors revised the paper according to my comments and now its structure is more solid. Paper describes the methodology in a more understandable way and quantifies the results more clearly. 

The structure of the reference style should be re-checked.

Author Response

(The authors gave the same response as above.)

Reviewer 3 Report

The authors addressed my comments properly. However, they should include their responses in the manuscript, either in the main text or in appendix, instead of only show them in the response file. This needs to be addressed before publication.

Author Response

(The authors gave the same response as above.)
